# Outcomes of Patients with Metastatic Castration-Resistant Prostate Cancer According to Somatic Damage DNA Repair Gene Alterations

Zoé Neviere [1,*], Elodie Coquan [1], Pierre-Emmanuel Brachet [1], Emeline Meriaux [1], Isabelle Bonnet [1], Sophie Krieger [2], Laurent Castéra [2], Dominique Vaur [2], Flavie Boulouard [2], Alexandra Leconte [3], Justine Lequesne [3], Anais Lelaidier [4], Agathe Ricou [2] and Florence Joly [1,3,5,6]

[1] Oncology Department, Centre François Baclesse, 14000 Caen, France; coque@baclesse.unicancer.fr (E.C.); (brape@baclesse.unicancer.fr (P.-E.B.); e.meriaux@baclesse.unicancer.fr (E.M.); i.bonnet@baclesse.unicancer.fr (I.B.); f.joly@baclesse.unicancer.fr (F.J.)
[2] Inserm U1245, Department of Cancer Biology and Genetics, Normandy Centre for Genomic and Personalized Medicine, François Baclesse Center, 14000 Caen, France; s.krieger@baclesse.unicancer.fr (S.K.); l.castera@baclesse.unicancer.fr (L.C.); d.vaur@baclesse.unicancer.fr (D.V.); f.boulouard@baclesse.unicancer.fr (F.B.); a.ricou@baclesse.unicancer.fr (A.R.)
[3] Clinical Reseach Department, Centre François Baclesse, 14000 Caen, France; a.leconte@baclesse.unicancer.fr (A.L.); j.lequesne@baclesse.unicancer.fr (J.L.)
[4] Data Processing Center, the North-West Canceropole, Centre François Baclesse, 14000 Caen, France; a.lelaidier@baclesse.unicancer.fr
[5] Department of Oncology, University Hospital of Caen, 14000 Caen, France
[6] UMR-S1077, Normandy University, Unicaen, Inserm U1086, Anticipe, 14000 Caen, France
* Correspondence: z.neviere@baclesse.unicancer.fr

**Abstract:** (1) Background: In literature, approximately 20% of mCRPC present somatic DNA damage repair (DDR) gene mutations, and their relationship with response to standard therapies in mCRPC is not well understood. The objective was to evaluate outcomes of mCRPC patients treated with standard therapies according to somatic DDR status. (2) Methods: Eighty-three patients were recruited at Caen Cancer Center (France). Progression-free survival (PFS) after first-line treatment was analyzed according to somatic DDR mutation as primary endpoint. PFS according to first exposure to taxane chemotherapy and PFS2 (time to second event of disease progression) depending on therapeutic sequences were also analyzed. (3) Results: Median first-line PFS was 9.7 months in 33 mutated patients and 8.4 months in 50 non-mutated patients ($p = 0.9$). PFS of first exposure to taxanes was 8.1 months in mutated patients and 5.7 months in non-mutated patients ($p = 0.32$) and significantly longer among patients with ATM/BRCA1/BRCA2 mutations compared to the others (10.6 months versus 5.5 months, $p = 0.04$). PFS2 was 16.5 months in mutated patients, whatever the sequence, and 11.7 months in non-mutated patients ($p = 0.07$). The mutated patients treated with chemotherapy followed by NHT had a long median PFS2 (49.8 months). (4) Conclusions: mCRPC patients with BRCA1/2 and ATM benefit from standard therapies, with a long response to taxanes.

**Keywords:** prostate cancer; molecular profile; homologous repair

## 1. Introduction

In the area of personalized medicine, the molecular characterization of tumors is becoming an integral feature of new therapeutic strategies, and some genetic alterations may be therapeutic targets [1,2]. Beyond germline mutations, somatic pathogenic variations acquired during the process of tumorigenesis can be found only within the tumor [3].

In prostate cancer, the DNA damage repair (DDR) pathway is one of the major genetic alterations with a potential therapeutic impact. The incidence of somatic alterations

in DDR pathways in prostate tumors is higher than that of germline alterations and varies from 19 to 31% in advanced prostate cancer and from 7.4 to 16.2% in germline mutations [1,4–8]. The major alterations concerned are *BRCA1/2* and *ATM* [4,9–12].

In patients with prostate cancer, germline pathogenic *BRCA1/2* variants are usually correlated with poor prognostic characteristics (aggressiveness, castration resistance, lymph node invasion and metastasis at diagnosis, and decreased overall survival) [13–17]. While the predictive impact of somatic *BRCA1/2* and other DDR mutations remains somewhat elusive [13–16], metastatic castration-resistant prostate cancer (mCRPC) patients with DDR mutations are known to have a response to PARP (poly(ADP-Ribose) polymerase) inhibitors [7,17–21].

The predictive impact of DDR gene germline mutations on the response to standard therapies (taxanes and/or new-generation hormone therapy [NHT]) was recently investigated in a first-line setting among mCRPC patients. However, results of the different studies remain conflicting about links between DDR mutations and survival outcomes after NHT and/or taxane treatments [9,11,14,16,22]. Annala et al. evaluated the predictive impact of somatic DDR alterations on circulating tumor DNA (ctDNA) in 115 mCRPC patients treated with first-line NHT [23]. They showed that defects in *BRCA2* and *ATM* were strongly associated with poor time to progression independently of clinical prognostic factors and circulating tumor DNA abundance ($p < 0.001$). Another retrospective study found worse PSA response rates (25%) in 53 mCRPC patients with somatic *BRCA2* mutations treated with docetaxel versus 71.1% in wild-type mCRPC patients ($p = 0.019$) [24].

Therefore, the predictive value of somatic alterations of DDR pathway genes in mCRPC patients treated with taxanes is still unclear, and sound data on progression-free survival are lacking. The objective of this study was to describe outcomes of mCRPC patients treated with taxanes and/or NHT according to their somatic DDR profile, determined with a large 65-gene panel.

## 2. Materials and Methods

This was an observational retrospective study conducted at the François Baclesse Center in Caen, France. Patients were screened between 1 January 2017 and 31 December 2018 during multidisciplinary meetings (Figure 1). Criteria of eligibility were all patients with mCRPC adenocarcinoma with evaluable lesions according to the PCWG3 and/or RECIST 1.1 criteria receiving a first-line treatment for mCRPC for at least 3 months, with tumor material available for somatic analysis. Patients may have received more than one line of mCRPC treatment after castration resistance. Patients with tumor types other than adenocarcinoma, with World Health Organization (WHO) performance status <2, or in whom the tumor material was insufficient or unavailable for somatic analysis were excluded.

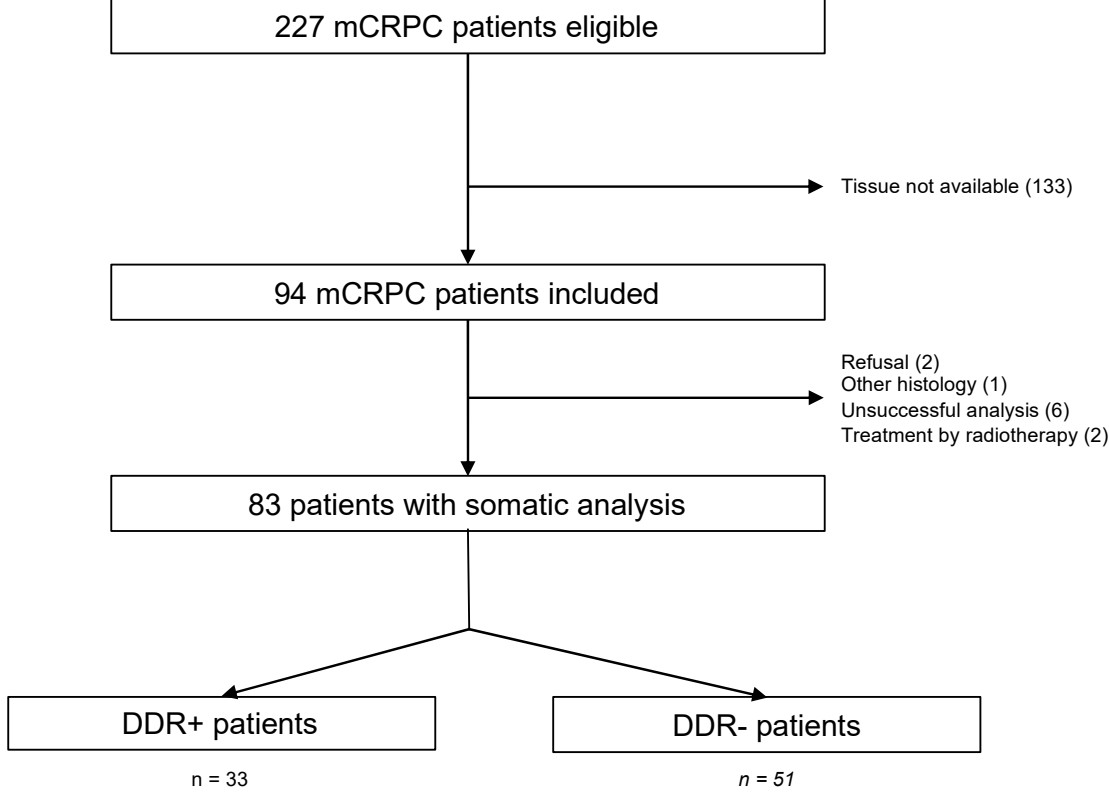

**Figure 1.** Flow chart of selection of patients. DDR+: mutated patients; DDR−: non-mutated patients; mCRPC: metastatic castration-resistant prostate cancer; *n*: number of patients.

Data were collected between 1 September 2018 and 15 March 2019 from patients' medical files at the François Baclesse Center. Their characteristics at the initial diagnosis (age at diagnosis, initial PSA, initial TNM, body mass index (BMI), Gleason score, diagnostic modes, different therapeutic lines, best response to castration resistance lines, progression dates after different lines and date of death or last follow-up) were collected. Monitoring ended on 15 March 2019.

Somatic analyses were performed on DNA extracted from the initial biopsy or surgical excision in the Laboratory of Biology and Genetics at the François Baclesse Center. The procedure and the panel are described in Table S1. Only likely pathogenic and pathogenic variations have been considered.

The primary endpoint was progression-free survival (mCRPC-PFS) after first-line castration-resistant treatment, defined as the date of the beginning of castration-resistant first-line treatment and the date of confirmed progression (biochemical as defined by the French Association of Urology and/or radiologic progression according to PCWG3 and/or RECIST 1.1 criteria). The secondary endpoints were PFS of the first mCRPC treatment to taxanes or to first-line hormonal therapy [NHT] during the first two lines of castration-resistant treatment, PFS2 (i.e., the time between the date of initiation of the first line of treatment for mCRPC and the date of progression with the second line of treatment), and overall survival (OS) (calculated from the beginning of the first line of treatment for mCRPC to the date of death or last follow-up). PFS2 was also evaluated according to the therapeutic sequence (NHT treatment for first-line castration resistance followed by taxane chemotherapy (HCS) or first-line chemotherapy followed by second-line NHT (CHS)).

Comparisons were made between patients with the DDR mutation (DDR+) and those without (DDR−). Analysis also concerned the group of patients with *BRCA1*, *BRCA2*, and/or *ATM* gene alterations (corresponding to the most frequent and already evaluated mutations).

Survival rates were estimated by the Kaplan Meier method. The log-rank test was used to determine factors associated with survival data. The link between the different factors and the molecular data obtained by sequencing the prostate tumors was measured by the Chi2 test in the event of qualitative variables (or Fisher test if necessary) and by the Student test in the event of quantitative variables (or the non-parametric Kruskal Wallis test if the data were not Gaussian). The significance threshold was set at 5% for each statistical analysis and confidence interval.

The Northwest Data Center (CTD-CNO) is acknowledged for managing the data. It is supported by grants from the French National League Against Cancer (LNC) and the French National Cancer Institute (INCa).

The study was approved by the institutional review board. It was conducted in compliance with the French Research Standard MR-004 "Research not involving human participants" (compliance commitment to MR-004 for the Centre François Baclesse n°2214228 v.0, dated from 07/03/2019). All data have been processed anonymously.

## 3. Results

### 3.1. Patient Selection

Two hundred and twenty-seven patients with mCRPC were eligible; 94 were included, and 83 patients were finally analyzed. Data regarding treatment group and prostate somatic DDR gene alteration are shown in Figure 1.

### 3.2. Clinical Characteristics

Table 1 summarizes the clinical characteristics of mCRPC patients according to DDR mutations. Median age was 69.5 years. Thirty-three patients (39.8%) presented the somatic DDR mutation (they represent the DDR+ group). There was no statistical difference between the two groups concerning clinical characteristics. Median follow-up since the date of castration resistance of the 83 patients with somatic analysis was 15.3 months [0.48–83.2].

**Table 1.** Clinical outcomes of analyzed patients according to somatic DDR+ versus DDR− alterations BMI: body mass index; DDR+: mutated patients; DDR−: non-mutated patients; n: number of patients; PSA: prostate specific antigen; NHT: new-generation hormonotherapy; PSA: prostate specific antigen.

| | **Total** | | **DDR+** | | **DDR−** | | *p* |
|---|---|---|---|---|---|---|---|
| Age(years) | 69.5 | (23–82) | 70 | (65–76) | 69.5 | (55–82) | 0.43 |
| ECOG | | | | | | | 0.21 |
| 0 | 34 | (44%) | 15 | (48%) | 19 | (41%) | |
| 1 | 38 | (49%) | 12 | (39%) | 26 | (55%) | NA |
| 2 | 6 | (8%) | 4 | (13%) | 2 | (4%) | NA |
| BMI | 27.5 | (23–38) | 27.5 | (25–31) | 27.5 | (23–36) | 0.71 |
| Previous Treatments | | | | | | | |
| Surgery | 22 | (27%) | 10 | (30%) | 12 | (24%) | 0.74 |
| Chemotherapy | 14 | (17%) | 6 | (18%) | 8 | (16%) | 1 |
| Radiotherapy | 45 | (54%) | 19 | (58%) | 26 | (52%) | 0.8 |
| First-line Treatment | | | | | | | 0.61 |
| NHT | 64 | (77%) | 14 | (73%) | 40 | (80%) | |
| Taxanes | 19 | (23%) | 9 | (27%) | 10 | (20%) | NA |
| Gleason | | | | | | | 0.47 |
| 5 to 7 | 35 | (43%) | 12 | (36%) | 23 | (47%) | |
| 8 to 10 | 47 | (57%) | 21 | (67%) | 26 | (53%) | NA |
| TNM | | | | | | | 0.85 |
| T1/2 | 42 | (51%) | 7 | (21%) | 14 | (28%) | |
| T3/4 | 41 | (49%) | 22 | (67%) | 30 | (60%) | NA |

| | | | | | | | |
|---|---|---|---|---|---|---|---|
| Tx | 10 | (12%) | 4 | (12%) | 6 | (12%) | |
| N1+ | 25 | (30%) | 8 | (24%) | 17 | (34%) | 0.55 |
| N0 | 16 | (19%) | 6 | (18%) | 10 | (20%) | NA |
| Nx | 42 | (51%) | 19 | (58%) | 23 | (46%) | NA |
| M1 | 43 | (52%) | 15 | (46%) | 28 | (56%) | 0.47 |
| M0/Mx | 40 | (48%) | 18 | (54%) | 22 | (44%) | NA |
| Initial Pas | 28.8 | (1–5500) | 28.8 | (9.7–60) | 27.6 | (10–232) | 0.32 |
| Diagnostic Modes | | | | | | | 0.73 |
| Symptoms | 49 | (62)% | 18 | (58%) | 31 | (65%) | |
| Individual screening | 30 | (38%) | 13 | (42%) | 17 | (35%) | NA |
| Durtion of Hormonosensitivity (years) | 2.07 | (0.4–18.1) | 2.14 | (0.5–18.1) | 1.92 | (0.4–13.9) | 0.5 |
| Time Before Metastasis (years) | 0.04 | (0–13.8) | 0.92 | (0–13.8) | 0.02 | (0–12.1) | 0.07 |

In first-line treatment, 64 mCRPC patients were treated with NHT and 19 with taxanes. Fifty-three patients (64%) received a second-line treatment: 31 patients received taxanes, and 22 patients received NHT second-line mCRPC. Forty-seven (57%) patients received at least one line of taxanes in the first two lines of treatment, and 74 patients received at least one line of NHT (89%). Ten mCRPC patients received chemotherapy followed by NHT, and 28 patients received NHT followed by taxanes. Fourteen patients had received chemotherapy before resistance to castration, and one of them had received taxanes as first-line mCRPC treatment.

Clinical characteristics of the 83 mCRPC patients analyzed and the 136 mCRPC patients excluded are shown in Supplementary Materials (Table S2). Patients included had more aggressive parameters, with more node invasion at diagnosis (30% vs. 17%, $p = 0.001$), higher Gleason score (57% of Gleason 8–10 vs. 39%, $p = 0.013$), and shorter time to metastasis (25 months vs. 0.5 months, $p = 0.05$). They also received more previous loco-regional radiotherapy ($p \leq 0.004$) and first-line taxanes for hormonosensitive disease ($p = 0.028$) and had a shorter median duration of first-line hormonosensitivity ($p = 0.0005$). Prognostic factors were similar between patients treated with taxanes and those with NHT as first line (Table S3).

### 3.3. Molecular Characteristics of Tumors

Thirty-three (39.7%) patients had at least one DDR alteration. Alterations concerned different genes: 10 patients presented alterations of the ATM gene (12%), 5 BRCA2 (6%), 4 CHEK2 (4.8%), 3 CDK12 and FANCG (3.6% for each gene), 2 MRE11A and PALB2 (2.4% for each gene), 1 BLM, 1 BRCA1, 1 CHEK1, 1 FANCF, 1 FANCI, 1 FANCM, and 1 MDC1 (1.2% for each gene). The subgroup of ATM/BRCA1/BRCA2 patients represented 19.2% of patients and 48.5% of mutated patients. Five samples (6%) had at least two somatic alterations, and one patient had a tumor with four somatic alterations including three pathogenic variants and one likely pathogenic variant. Another alteration was reported that was a likely pathogenic variant and localized on the ATM gene. The mutations are shown in Table 2 (Figure S1: complementary results).

**Table 2.** Pathogenic or likely pathogenic variants identified on prostatic tumor somatic analysis among mCRPC cohort.

| Patient | Gene | Alteration | Protein | Function | Types |
|---|---|---|---|---|---|
| 1 | *ATM* | 5188C > T | ARG1730* | stop | Pathogenic |
| 2 | *CDK12* | 2068DEL | ALA690GLNFS*63 | frameshift | Pathogenic |
| 2 | *CDK12* | 3046C > T | GLN1016* | stop | Pathogenic |
| 8 | *ATM* | 4403T > A | VAL1468ASP | missense | Pathogenic |
| 9 | *firefox* | 1100DEL | THR307METFS*15 | frameshift | Pathogenic |
| 14 | *BRCA1* | 3741DEL | ALA1248LEUFS*16 | frameshift | Pathogenic |
| 15 | *MRE11A* | 571C > T | ARG191* | stop | Pathogenic |
| 16 | *ATM* | 5712DUP | SER1905ILEFS*25 | frameshift | Pathogenic |
| 17 | *BRCA2* | - | - | - | Pathogenic |
| 18 | *CDK12* | 3566_3575DEL | LEU1189GLNFS*23 | frameshift | Pathogenic |
| 19 | *PALB2* | 658_659DEL | SER220CYSFS*14 | frameshift | Pathogenic |
| 25 | *BRCA2* | 5909C > A | SER1970* | stop | Pathogenic |
| 27 | *CHEK2* | 1100DEL | THR367METFS*15 | frameshift | Pathogenic |
| 28 | *MDC1* | 907DEL | VAL303TRPFS*45 | frameshift | Pathogenic |
| 30 | *ATM* | 9022C > T | ARG3008CYS | missense | Pathogenic |
| 30 | *ATM* | 8096C > T | PRO2699LEU | missense | Pathogenic |
| 34 | *ATM* | 5293_5302DEL | GLN1765GLUFS*8 | frameshift | Pathogenic |
| 38 | *ATM* | 8759_8772DEL | ILE2920ARGFS18* | frameshift | Pathogenic |
| 39 | *BLM* | 1701G > A | TRP567* | stop | Pathogenic |
| 40 | *CHEK2* | - | TYR370CYS | missense | Pathogenic |
| 51 | *CHEK1* | 783DEL | ASP262ILEFS*42 | frameshift | Pathogenic |
| 53 | *FANCM* | 1827T > G | TYR609* | stop | Pathogenic |
| 53 | *CDK12* | 467_470DEL | GLU156GLYFS*10 | frameshift | Pathogenic |
| 54 | *FANCG* | 1183_1192DEL | GLU375TRPFS* | frameshift | Pathogenic |
| 56 | *FANCF* | 1087C > T | GLN363* | stop | Pathogenic |
| 59 | *ATM* | 5818G > T | GLU1940* | stop | Pathogenic |
| 60 | *BRCA2* | 1597DEL | THR533LEUFS*25 | frameshift | Pathogenic |
| 62 | *MRE11A* | 1331_1332DEL | VAL444ALAFS*2 | frameshift | Pathogenic |
| 63 | *BRCA2* | C.1813DEL | ILE605TYRFS*9 | frameshift | Pathogenic |
| 68 | *FANCG* | 572T > G | LEU191* | stop | Pathogenic |
| 75 | *ATM* | 7306A>G | ARG2436GLY | missense | Pathogenic |
| 76 | *BRCA2* | 5073DUP | TRP1692METFS*3 | frameshift | Pathogenic |
| 76 | *FANCI* | 3184C>T | GLN1082* | stop | Pathogenic |
| 76 | *FANCG* | 1143G>C | ARG381SER | missense | Likely pathogenic |
| 76 | *BRCA2* | 7307DEL | ASN2436THRFS*33 | frameshift | Pathogenic |
| 78 | *ATM* | 901G>A | GLY301SER | faux sens | Pathogenic |
| 81 | *ATM* | 7031G>A | TRP2344* | stop | Likely pathogenic |
| 81 | *SMARCA2* | 4369C>T | ARG1457CYS | missense | Pathogenic |
| 82 | *PALB2* | 2850DEL | SER951LEUFS*11 | frameshift | Pathogenic |
| 83 | *CHEK2* | 1116_1117DEINSTG | LYS373GLU | missense | Pathogenic |

*3.4. PFS*

3.4.1. First-Line PFS

The first-line median PFS of mCRPC patients was 9.7 months. No difference was observed between DDR+ and DDR− patients (9.8 months versus 8.4 months; $p = 0.91$; Figure 2A). The PFS of the 16 patients with an *ATM/BRCA1/BRCA2* mutation was 14.4 months versus 8.3 months for the other patients ($p = 0.24$; Figure 2B).

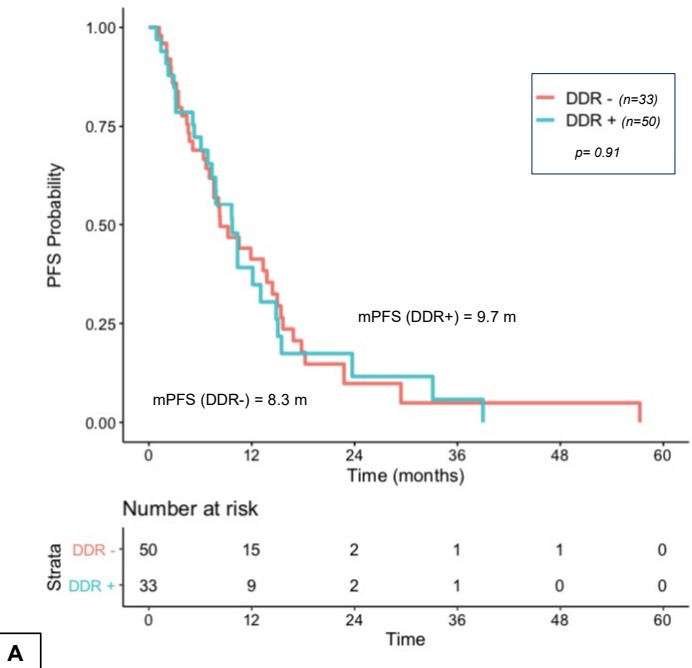

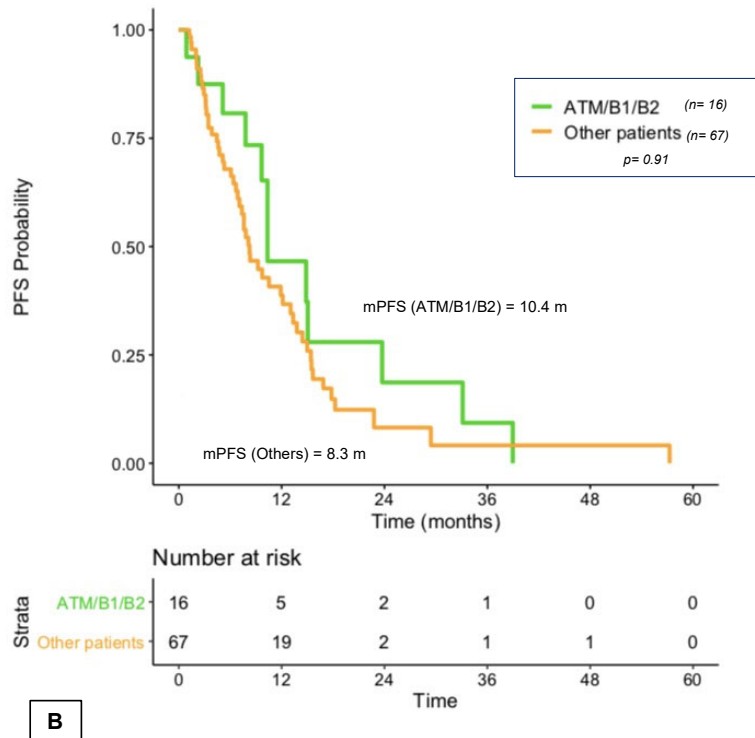

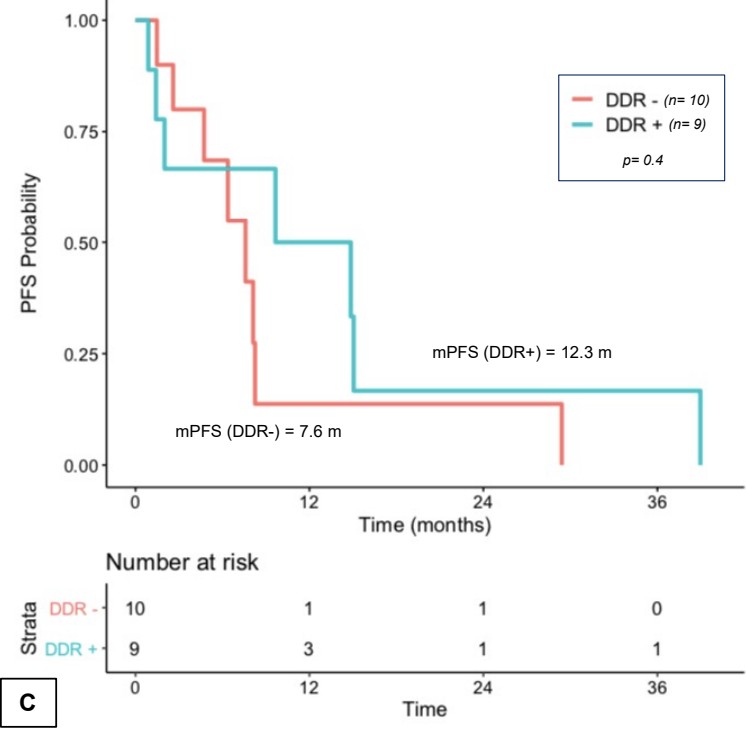

**Figure 2.** First-line treatment mCRPC PFS according to somatic DDR+ versus DDR− altera-tions (**A**), according to ATM/B1/B2-mutated patient versus other patients (**B**). DDR+: mutated patients; DDR−: non-mutated patients; m: months; mCRPC: metastatic castration-resistant prostate cancer; PFS: progression-free survival; n: number of patients. (**C**) First-line treatment mCRPC PFS according to somatic DDR+ versus DDR− alterations among patients who received taxanes. DDR+: mutated patients, DDR−: non-mutated patients; m: months; mCRPC: metastatic castration-re-sistant prostate cancer; mPFS: median progression-free survival; *n*: number of patients.

For patients treated by taxanes in first line, median PFS of the 9 DDR+ mCRPC pa-tients was 12.3 months, compared to 7.6 months in the 10 DDR− patients (*p* = 0.4; Figure 2C). The PFS of the 6 *ATM/BRCA1/BRCA2* patients treated in first line with taxanes was 14.9 months, compared to 6.4 months for the 13 other patients treated with taxanes with another or no mutation (*p* = 0.11).

For patients treated by NHT in first line, median PFS was 9.8 months for the 24 DDR+ and 12 months for 40 DDR− patients (*p* = 0.68; Figure S2). In patients treated by NHT, median PFS of the 10 *ATM/BRCA1/BRCA2* was 10.4 months versus 8.3 months for the other patients (*p* = 0.43). No statistical difference between mCRPC first-line PFS of the 6 patients with the somatic *BRCA1/BRCA2* mutation and the 10 patients with *ATM* muta-tions was observed (respectively 10.3 months and 10.3 months; *p* = 0.69).

3.4.2. PFS with First Exposure to Taxanes and NHT

Among the 47 patients who received at least one line of taxanes in the first two lines of treatment, the PFS with first exposure to taxanes was 8.1 months in DDR+ patients and 5.7 months in DDR− patients (*p* = 0.31; Figure 3A). It was 10.6 months for the 9 *ATM/BRCA1/BRCA2*-mutated patients versus 5.5 months for the other patients (*p* = 0.04; Figure 3B). Median PFS in the 6 patients with the somatic *ATM* mutation was 9.7 months versus 15.1 months mPFS in the 3 patients with somatic *BRCA1/2* mutations (*p* = 0.14).

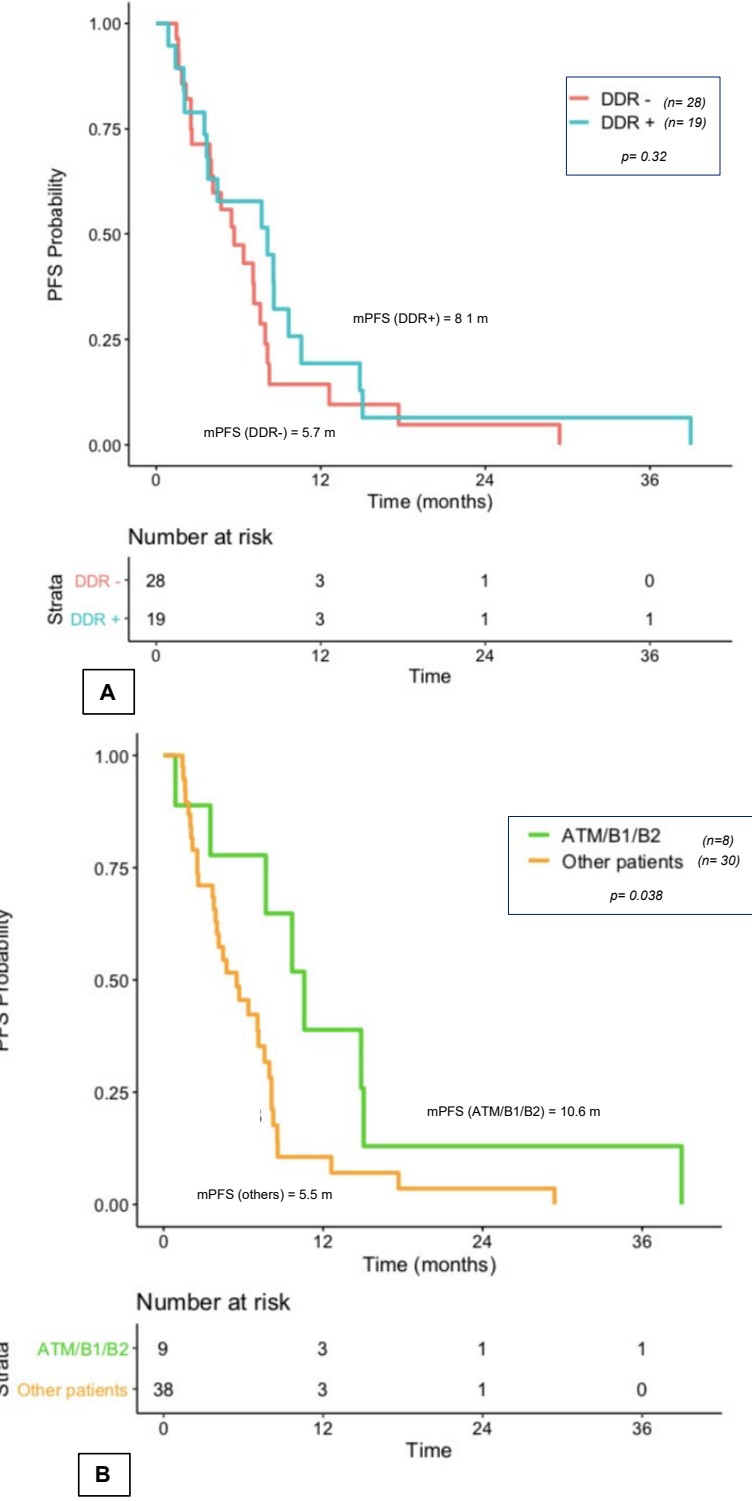

**Figure 3.** (**A**) First exposure to taxanes among mCRPC patients in first two lines according to somatic DDR+ versus DDR− alterations (**A**) according to ATM/B1/B2-mutated patients versus other patients. (**B**). DDR+: mutated patients, DDR−: non-mutated patients; m: months; mCRPC: metastatic castration-resistant prostate cancer; mPFS: median progression-free survival; *n*: number of patients.

For the 74 patients who received at least one line of NHT, first exposure PFS was similar in the two groups (9.7 months for DDR+ vs. 8.3 months for DDR−; $p$ = 0.73; Figure S3A). In this group, PFS was 10.4 months in *ATM/BRCA1/BRCA2*-mutated patients versus 7.8 months for the other patients ($p$ = 0.22; Figure S3B). Median PFS in the 7 patients with the somatic *ATM* mutation was 23.7 months versus 9 months mPFS in the 6 patients with somatic *BRCA1/2* mutations ($p$ = 0.056).

### 3.5. PFS2

Among all patients who received at least two lines of mCRPC treatment, PFS2 of DDR+ patients was 16.7 months versus 12.6 months for DDR− patients ($p$ = 0.88; Figure S4A). PFS2 of *ATM/BRCA1/BRCA2*-mutated patients was 18.2 months versus 12.6 months for the others ($p$ = 0.11; Figure S4B).

Among the 38 patients who received NHT and chemotherapy during the first two lines for mCRPC, median PFS2 of the 10 patients who received chemotherapy followed by NHT was 11.7 months, and median PFS2 of the 28 mCRPC patients who received NHT followed by taxanes was 13.2 months ($p$ = 0.56; Figure 4A). PFS2 of the 3 *ATM/BRCA1/BRCA2*-mutated patients treated with the taxane-NHT sequence was 49.8 months. PFS2 of DDR+ patients was 16.5 months, whatever the sequence, versus 11.7 months for DDR− patients ($p$ = 0.07; Figure 4B). In this chemotherapy and NHT group, the 6 *ATM/BRCA1/BRCA2*-mutated patients had a much longer PFS2 compared to patients with another or no mutation (median PFS2 of 35.7 months versus 11.7 months; $p$ = 0.004). In *ATM/BRCA1/BRCA2*-mutated patients treated by taxane and then the NHT sequence, PFS2 was particularly long (median PFS = 49.8 months) vs. 27.4 months for the reverse sequence ($p$ = 0.19). No statistical difference was observed between PFS2 of the 4 patients with the somatic *BRCA1/BRCA2* mutation and the 6 patients with the *ATM* mutation (respectively, 16.5 months and 22.8 months; $p$ = 0.7).

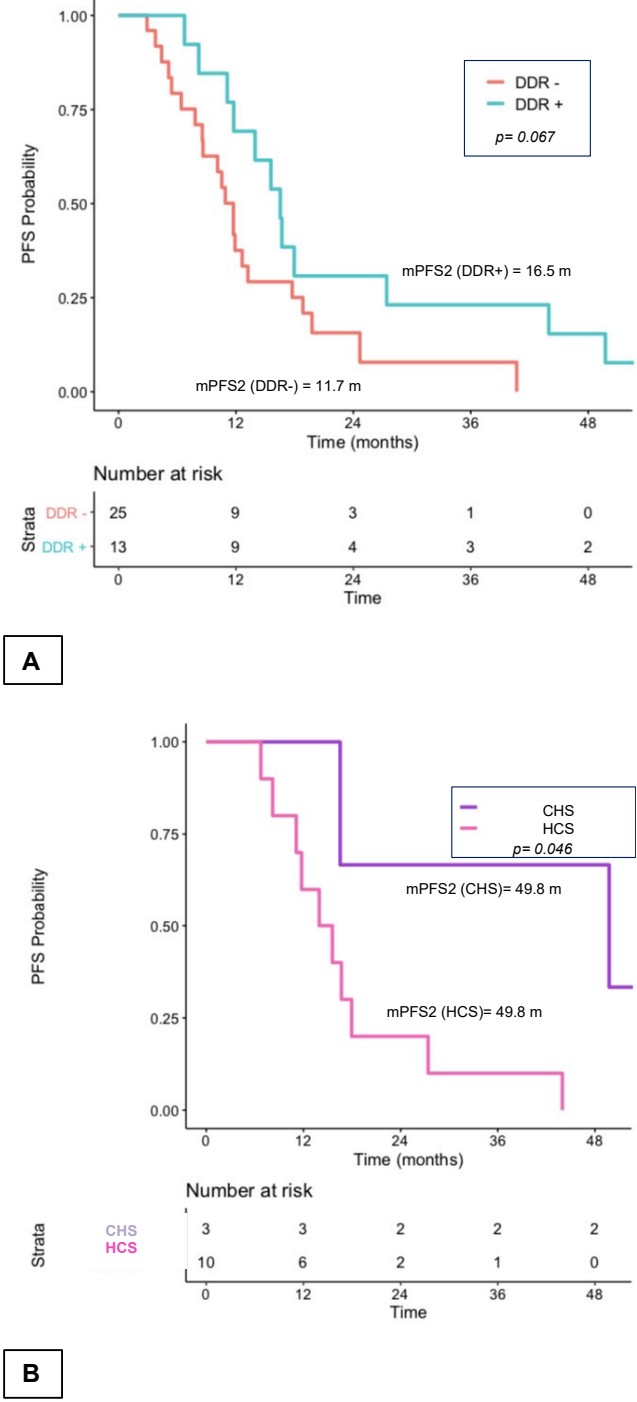

**Figure 4.** PFS2 according to somatic DDR+ versus DDR− alterations and among patients who received only CHS or HCS (**A**) and PFS2 among mutated patients who received CHS or HCS according to sequence. (**B**) CHS: chemotherapy followed by NHT sequence; DDR+: mutated patients; DDR−: non-mutated patients; HCS: NHT followed by chemotherapy sequence; m: months; mCRPC: metastatic castration-resistant prostate cancer; median PFS: median progression-free survival; *n*: number of patients.

*3.6. Overall Survival*

Of the 83 patients, 35 patients died. Median OS was 2.2 years in DDR− group and was not reached in the DDR+ group (*p* = 0.39) or in the *ATM/BRCA1/BRCA2* group (*p* = 0.7) (Figure S5).

## 4. Discussion

In our study, patients with somatic mutations of *ATM/BRCA1/BRCA2* genes achieved longer PFS with standard mCRPC treatments than other patients. They seem to receive greater benefit from taxanes. Moreover, alterations of the different DDR genes do not have the same predictive value.

In this series, 40% of mCRPC patients presented a somatic DDR gene alteration. This rate is higher than those previously described in the literature among mCRPC patients, which range from 21% to 32% [1,7,18,25]. While most of those studies focused on a narrow gene panel including only two to 22 genes, we used a much larger panel of 69 genes. On the other hand, *ATM/BRCA1/BRCA2*-mutated patients represent 19% of our cohort, which is consistent with other studies.

Our molecular analysis could not determine whether alterations were mono- or bi-allelic, so the involvement of these alterations in carcinogenesis remains unknown. Some of these alterations might only be passenger mutations with little predictive significance. Moreover, we included patients with aggressive disease; they probably had several somatic mutations. In France, tumor samples are not required to be kept for more than 10 years, so patients with an initial diagnosis dating back over 10 years were usually excluded from our study because of the lack of availability of tumor samples. The patients who were included, therefore, had more aggressive tumors with a short duration of hormonosensitivity (Table S2). This is consistent with the fact that patients with germline *BRCA1/BRCA2* mutations generally develop more aggressive tumors [13,14].

Considering the whole population, outcomes of the DDR+ group and DDR− group were not different, whatever the first-line setting and the sequencing of treatments. This is the first report evaluating first-line mCRPC PFS according to somatic DDR mutations, whatever the treatment, even though the few studies reporting PFS according to heterogenous germline mutations reported the same results [11,22]. Most of the prostatic somatic DDR mutations concerned the *BRCA1/2* and *ATM* genes [1,5]. *BRCA1/BRCA2*/ATM mutations were the first reported molecular alterations conferring sensitivity to PARP inhibitors in prostate cancer and are the most widely studied germline and somatic mutations in this setting [11,26].

To be able to compare our results with previous studies, we also focused on the subgroup of patients with *ATM/BRCA1/BRCA2* mutations. Our study is the first to report data on PFS after mCRPC taxane treatment according to the presence of somatic DDR and *ATM/BRCA1/BRCA2* alterations. Outcomes were better for patients with *ATM/BRCA1/BRCA2* mutations treated with taxanes. These patients with first-taxane exposure had a two-fold longer PFS than those with another or no somatic mutation. Median PFS2 for these mutated patients was also particularly high (around 4 years) with the taxane-NHT sequence. Other studies reported different PFS of mCRPC patients treated with taxanes, but the populations were screened on the basis of germline alterations. Annala *et al.* did not find any significant difference in PSA-PFS among eight DDR− mutated and 18 non-mutated patients treated by taxanes. Likewise, Mateo et al. did not find any difference in PFS between 44 DDR− mutated patients versus 238 non-DDR− mutated ones treated with taxanes, and there was no difference according to *BRCA2* mutations. In the study by Castro et al., PFS with first exposure to taxanes and PFS2 with taxanes followed by NHT in the subgroup of 14 *BRCA2*-mutated patients were shorter than those of patients with no germline *BRCA2* mutation [4]. However, it is difficult to draw any conclusions, due to the heterogeneity of those studies. Indeed, our series is a small retrospective singe-center cohort with a brief follow-up time. The other studies also included a limited number of

patients selected according to germline mutations and heterogenous panels of genes. In our study, PFS2 was longer when somatic *ATM/BRCA1/BRCA2*-mutated patients were treated by taxanes followed by NHT rather than vice versa. Finally, a small study explored this question of sequence and reported different results: PFS2 of the seven mCRPC patients with the germline *BRCA2* mutation who received taxanes followed by NHT was shorter than that of seven *BRCA2*-mutated patients who received NHT followed by taxanes [4]. However, the subgroup of patients was screened differently, i.e., germline BRCA2-mutated patients versus somatic *ATM/BRCA1/BRCA2* alterations in our study.

In our series, we did not find any difference between groups treated by NHT. Regarding PFS related to NHT, two other studies confirmed our finding, since they found no strong relationship between mutation and first exposure to NHT [4,11]. Two other studies found different results from ours but with conflicting conclusions. One found better PFS in patients with germline *BRCA2/ATM* mutations treated by NHT than in those without the *BRCA2/ATM* mutation (15.2 versus 10.8; *p* = 0.044) [22]. On the other hand, Annala *et al.* found shorter PFS in mutated patients treated with NHT in first line, first in a retrospective study and then in a prospective cohort exploring the predictive impact of *BRCA2* and *ATM* mutations identified in circulating tumor DNA [9,23]. Again, this difference in PFS is likely due to heterogenous gene panels, with either somatic, germline, or circulating tumoral DNA, small series, and follow-up that was too short.

It is difficult to compare these studies because of their heterogeneous populations and screening criteria. Moreover, the panels used were different, and so the predictive impact of the different DDR gene alterations was probably lessened.

In our study, the PFS of *ATM/BRCA1/BRCA2*-mutated patients and of those with other DDR mutations treated with taxanes or NHT was not similar. Patients with an *ATM/BRCA1/BRCA2* mutation had significantly longer PFS2 than those with other mutations when receiving standard treatments. PFS of first exposure to NHT was not statistically different between patients with *ATM* or *BRCA1/BRCA2* mutations, while patients with the *ATM* mutation seemed to have a longer PFS than those with *BRCA1/2* mutations. Contrary results were observed regarding first exposure to taxanes, where PFS was longer in patients with BRCA mutations than in those with ATM mutations. Alterations of the different DDR genes probably do not have the same predictive impact.

This issue has also received attention in patients treated with PARP inhibitors. Marshall et al. observed that PFS in mCRPC patients treated with olaparib with an *ATM* mutation was shorter than that in patients with the *BRCA1/BRCA2* mutation [18]. Gene mutations were germline and/or somatic. The PROFOUND trial compared olaparib to NHT in patients with *ATM/BRCA1/BRCA2* mutations and in those with other DDR mutations screened by a 15-gene panel. Patients with *ATM/BRCA1/BRCA2* mutations had a better PFS with olaparib than with NHT [17]. Exploratory results of PFS by type of mutation showed that *BRCA2*- and *RAD51B*-mutated patients tended to have better PFS than *ATM*- or *BRCA1*-mutated patients. In the TRITON2 trial, which evaluated response to rucaparib in DDR+ mCRPC patients, a limited number of radiographic and PSA responses was observed in patients with *ATM, CDK12,* or *CHEK2* gene alterations, whereas responses were observed in patients with alterations in other DDR genes, such as *PALB2, BRIP1, FANCA,* and *RAD51B*. These studies showed that responses differ according to the somatic DDR alteration [27]. Alteration of the different DDR genes seems to have an independent predictive value for PARP inhibitors and for standard therapies.

Although outcomes were not different between our DDR+ and DDR− patients, whatever the first mCRPC line of treatment setting and the sequencing of treatments, mCRPC patients with the *ATM/BRCA1/BRCA2* mutation benefited from standard therapies, with long responses to taxanes in the BRCA1/2 mutation group and to NHT in patients with the ATM mutation. This reinforces the idea that the predictive impact of the alterations of the different DDR genes varies according to the type of treatment and gene concerned. In the setting of mCRPC, the optimal therapeutic sequence remains elusive. If predictive biomarkers could be established for choosing one particular treatment over another and for

knowing the outcomes of standard treatment for the different DDR+ gene, this could help in selecting the best treatment sequence [28,29].

Because this study presents several limitations, such as monocentric and retrospective characteristics, small sample size, and heterogeneous population (prior treatment, metastasis at diagnosis…), new prospective studies with more homogenous patients would be needed to confirm these results.

## 5. Conclusions

Metastatic CRPC patients with the ATM/BRCA1/BRCA2 mutation benefit from standard therapies, with long responses to taxanes. The predictive impact of DDR genes is probably dependent on the gene and the systemic treatment. Future studies are needed to confirm these findings.

In the area of PARP inhibitors, taxane before or after PARP inhibitors should be discussed.

**Supplementary Materials:** The following supporting information can be downloaded at: www.mdpi.com/xxx/s1, Table S1—Lists of genes in the panel and methodology. Table S2—Outcomes of analyzed versus excluded patients. Table S3—Clinical outcomes of analyzed patients according to treatments (NHT versus Taxanes). Figure S1—Annex of molecular results. Figure S2—First-line treatment mCRPC PFS according to somatic DDR+ *versus* DDR− alterations among patients who received NHT. Figure S3—First exposure to NHT among mCRPC patients in first two lines according to somatic DDR+ versus DDR− alterations and (A) according to ATM/B1/B2-mutated patients versus other patients (B). Figure S4—PFS2 according to somatic DDR+ versus DDR− alterations (A) according to ATM/B1/B2-mutated patients versus other patients (B), according to somatic DDR+ versus DDR−. Figure S5—OS according to somatic DDR+ versus DDR− alterations (A) and according to ATM/B1/B2-mutated patient versus the other patients (B).

**Author Contributions:** The landscape of personalized medicine is in a large and rapid mutation with molecular profiling. In this article, we propose to explore outcomes of patients with metastatic castration-resistant prostate cancer (mCRPC) according to somatic DNA damage repair genes alterations and standard mCRPC therapies. This is the first paper that proposes outcomes among a mCRPC cohort selected with a large genetic panel screening somatic alterations and standard therapies. Z.N. and F.J. wrote the manuscript and devised the study concept and design. P.-E.B., E.C., E.M., and I.B. edited the paper. J.L. was responsible for overseeing the statistical section. A.L. (Anais Lelaidier) contributed to data collection. A.L. (Alexandra Leconte) took care declarations. A.R., D.V., F.B., L.C., and S.K. carried out somatic analysis. All authors reviewed the paper. All authors have read and agreed to the published version of the manuscript.

**Funding:** This research received no external funding.

**Institutional Review Board Statement:** The PROSOTAX retrospective observatory is conducted in accordance with the amended law of 6/01/1978 relating to information technology, files, and freedoms, the EU regulation n 2016/679 relating to data protection ("RGPD"), and the law n 2018-493 of 20/06/2018 relating to personal data protection. The protocol was approved by the Centre François Baclesse, the study promoter. All patients are given global information on potential use of their data registered during their management in the institution; in addition, all patients are specifically informed about this study. Patients are free to express their opposition to the use of their data at any time. This procedure is in line with the French Research Standard MR-004 "Research not involving human participants". The study was approved by the local (internal) IRB from our institution "Centre François Baclesse". This procedure is in line with the French Research Standard MR-004 "Research not involving human participants".

**Informed Consent Statement:** All patients are given global information on potential use of their data registered during their management in the institution; in addition, all patients are specifically informed about this study. Patients are free to express their opposition to the use of their data at any time.

**Data Availability Statement:** The datasets used and/or analyzed during the current study are available from the corresponding author on reasonable request.

**Acknowledgments:** We are grateful to all of the patients and their caregivers. We acknowledge the Data Processing Centre (DPC) of the Northwest Canceropole (Centre de Traitement des Données du Cancéropôle Nord-Ouest) in charge of data management. The investigators are also thanked.

**Conflicts of Interest:** The authors declare no conflict of interest.

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
