# Peer review of "Outcomes of Patients with Metastatic Castration-Resistant Prostate Cancer According to Somatic Damage DNA Repair Gene Alterations"

_curroncol, doi:10.3390/curroncol29040226_

Round 1

Reviewer 1 Report

The authors present a study of the outcomes of patients with metastatic castration-resistant prostate cancer focusing on the impact of somatic mutation of damaged DNA repair genes. The authors should include a complete description of the methods used in the manuscript before I can give an appropriate evaluation.

I list here below my comments:

ABSTRACT

1. In the abstract, the authors did not introduce where the study was conducted, and the number of patients recruited. It is not clear if “More than 20% of mCRPC present somatic DNA damage repair gene mutations (DDR)” is referred to the study cohort or is referred to information reported in the literature. However, in this study, the authors reported (Figure 1) that 33 of 84 patients carry DDR gene mutations. The author reported that 40% of patients of their cohort are DDR+

2. Acronyms should be put in the right position in the text. Example: please, change “DNA damage repair gene mutations (DDR)” in “DNA damage repair (DDR) gene mutations

3. The author did not highlight the experiments performed. How many DDR genes were analyzed? Which technique was used?

MAIN TEXT

4. Figure 1. The sum of the patients DDR+ (n=33) and DDR- (n=51) differs from the total number reported (n=83)

5. There is no description of the experimental procedure used to identify the mutation. On page 3, line 85, the authors refer to the Table S1. In Table S1, there is no information about the experimental procedure used.

6. How the mutations on DDR genes are classified? How do you identify the patients as DDR+?

7. Table 1: I suggest removing the columns “n”. It is not clear the meaning of square brackets: CI95%, range, or interquartile range? Please, check if the information reported are corrected. The initial PSA for all patients ranges between 1 and 5500, while for DDR+ patients and DDR- patients ranges respectively from 9.7 to 60 and from 10 to 232.

8. Please, summarize the results of 12 survival curve plots in a single table. Show in the manuscript only the most relevant survival curves, the others can go in Supplementary material

Author Response

Dear reviewer,

Thank you for your constructive remarks.

You find all point-by-point responses below.

Best regards,

Dr Zoé NEVIERE

ABSTRACT

  1. In the abstract, the authors did not introduce where the study was conducted, and the number of patients recruited.

>> We have added “ 83 patients were recruited at Caen Cancer Center (France).”

It is not clear if “More than 20% of mCRPC present somatic DNA damage repair gene mutations (DDR)” is referred to the study cohort or is referred to information reported in the literature. However, in this study, the authors reported (Figure 1) that 33 of 84 patients carry DDR gene mutations. The author reported that 40% of patients of their cohort are DDR+

>> We have specified “in literature “for this report. Indeed, in our cohort, we show 37,5% of somatic DDR mutations.

  1. Acronyms should be put in the right position in the text. Example: please, change “DNA damage repair gene mutations (DDR)” in “DNA damage repair (DDR) gene mutations

>> We have modified the position as suggested.

  1. The author did not highlight the experiments performed. How many DDR genes were analyzed? Which technique was used?

>> We have chosen to specify it in methodology (abstract was to short.). 69 genes have been analyzed by NextGenerationSequencing.

MAIN TEXT

  1. Figure 1. The sum of the patients DDR+ (n=33) and DDR- (n=51) differs from the total number reported (n=83)

>> We have changed the number of patients in DDR- group to 50. It was a mistake.

  1. There is no description of the experimental procedure used to identify the mutation. On page 3, line 85, the authors refer to the Table S1. In Table S1, there is no information about the experimental procedure used.

>> We have specified in methodology in Table S3, modified as Table S1 as referee in the manuscript. We have used a 69 DDR-gene panel, to seuquence by NGS tumor. Analysis were realized in our Laboratory of biology and genetics.

  1. How the mutations on DDR genes are classified? How do you identify the patients as DDR+?

>> The classification was made by our Laboratory of Biology and Genetics at the François Baclesse Center, Inserm U1245, national referent for genetic analysis. Only likely pathogenic and pathogenic variations have been considered. We have added this precision in metholodolgy.

  1. Table 1: I suggest removing the columns “n”. It is not clear the meaning of square brackets: CI95%, range, or interquartile range? Please, check if the information reported are corrected. The initial PSA for all patients ranges between 1 and 5500, while for DDR+ patients and DDR- patients ranges respectively from 9.7 to 60 and from 10 to 232.

>> We have removed the columns “n”. The square brackets correspond to the first and the third interquartile ranges.

  1. Please, summarize the results of 12 survival curve plots in a single table. Show in the manuscript only the most relevant survival curves, the others can go in Supplementary material

>> You can find the synthesis of results in the next table. We have choosen to keep Figure 2A A/B/C Figure 3A A/B and Figure 4 C/D. We have modified these in the manuscript, and added others figures to supplementary data.

Corresponding Figures

Patient groups

n

Median PFS (months)

p

First-line treatment mCRPC PFS according to somatic DDR+ versus DDR- alterations

Figure 2A

A

DDR-

50

8,3

0,91

DDR+

33

9,7

First-line treatment mCRPC PFS according to ATM/B1/B2 mutated-patient versus other patients

Figure 2A

B

ATM/B1/B2

16

10,4

0,24

Other patients

67

8,3

First-line treatment mCRPC PFS according to somatic DDR+ versus DDR- alterations among patients who received taxanes

Figure 2B

C

DDR-

10

7,6

0,4

DDR+

9

12,3

First-line treatment mCRPC PFS according to somatic DDR+ versus DDR- alterations among patients who received NHT

Figure 2B

D

DDR-

40

11,9

0,68

DDR+

24

9,7

First exposure to taxanes among mCRPC patients in first two lines according to somatic DDR+ versus DDR- alterations

Figure 3A

A

DDR-

28

5,7

0,31

DDR+

19

8,1

First exposure to taxanes among mCRPC patients in first two lines according to ATM/B1/B2 mutated patients versus other patients

Figure 3A

B

ATM/B1/B2

9

10,6

0,038

Other patients

38

5,5

First exposure to NGHT among mCRPC patients in first two lines according to somatic DDR+ versus DDR- alterations

Figure 3B

C

DDR-

47

8,3

0,73

DDR+

27

9,7

First exposure to NGHT among mCRPC patients in first two lines according to ATM/B1/B2 mutated patients versus other patients

Figure 3B

D

ATM/B1/B2

13

10,4

0,22

Other patients

61

7,8

PFS2 according to somatic DDR+ versus DDR- alterations

Figure 4

A

DDR-

36

12,6

0,88

DDR+

21

16,7

PFS2 according to ATM/B1/B2-mutated patients versus other patients

Figure 4

B

ATM/B1/B2

10

18,2

0,11

Other patients

47

12,6

PFS2 according to somatic DDR+ versus DDR- alterations and among patients who received only CHS or HCS

Figure 4

C

DDR-

25

11,7

0,067

DDR+

13

16,5

PFS2 among mutated patients who received CHS or HCS according to sequence

Figure 4

D

CHS

3

49,8

0,046

HCS

10

14,8

Reviewer 2 Report

Reviewer comments regarding:  Outcomes of patients with metastatic castration-resistant prostate cancer according to somatic damage DNA repair gene alterations.

Strengths:

  • This is a retrospective study that examines outcomes of standard of care therapies, particularly taxanes, and androgen receptor pathway inhibitors (ARPIs) based on the presence or absence of DDR gene alterations on somatic testing.
  • This is a clear and well written manuscript.
  • This represents and important addition to the medical literature as the clinical behaviour of this patient population is not yet well defined, and prior studies offer conflicting results, as the authors have pointed out.

Suggestions for revisions:

  1. Line 46: “patients with DDR mutations are known to have a high response to PARP (Poly(ADP Ribose) Polymerase) inhibitors” - I would soften this statement. There is strong evidence that BRCA2 has high response rates to PARP inhibitors, but the other DDR genes appear to be far less responsive.
  2. Minor comment: The use of the term new-generation hormone therapy (NGHT) is not consistent with other authors in the field, and these agents are no longer “new”.  The most frequently used terms are “androgen receptor pathway inhibitors (ARPIs)” and “novel hormonal therapies (NHT)”.  Perhaps consider changing to one of these terms.
  3. Lines 50-52: “However, the debate remains open 50 since mutations were linked to better or worse survival outcomes after NGHT and/or taxane treatments”. I find this sentence structure awkward.  I may suggest changing to say that studies have shown conflicting results with taxanes and/or ARPIs.
  4. Minor comment: Line 72 “WHO>2” Please define the “WHO” before using the acronym, and presumably this should read <2 (ie 0 or 1).
  5. Figure 1: Statement that 2 patients were excluded due to refusal.  Does this imply that the other patients provided consent?  If so I would include this in the methods, as this would strengthen the study.  I highlight this as it is not typical for retrospective studies to require patient consent, but is obviously ethically favourable if consent is obtained.
  6. Line 88: Please define what you mean by progression (ie biochemical and/or radiographic? and what criteria were used, ie PCWG3, RECIST, etc.).
  7. Line 122-123: “Thirty-three patients (39.8%) presented the DDR 122 mutation (DDR+)” Awkward sentence. Suggest changing to “Thirty-three patients (39.8%) presented the DDR 122 mutation (DDR+)”.
  8. Table 1: Please include units of measure in brackets beside each row title ex.  “Age (years)”.
  9. Table 1: Consider changing title “Duration of hormonosensitivity” to “time to castration resistance”.  Also, I assume this is measure in years?  See point above.
  10. Table 2: Suggest using accepted terminology of "pathogenic" and "likely pathogenic”
  11. Results section: For patients treated with taxanes, It is not clearly stated whether or not patients that received taxane chemotherapy in the mCSPC state are including in this analysis.  This is essential to know as one would expect lower response rates and shorter time to progression with retreatment.  It would be preferable to exclude these patients from the analysis (or do another analysis) to allow a more pure look at outcome from initial taxane use in this patient population.
  12. Lines 231-235: “In this chemotherapy and NGHT group, the 6 ATM/BRCA1/BRCA2 -231 mutated patients had a much longer PFS2 compared to patients with another or no mutation (median PFS2 of 35.7 months versus 7 months; p = 0.004). In ATM/BRCA1/BRCA2 mutated patients treated by taxane and then the NGHT sequence, PFS2 was particularly long (median PFS = 49.8 months) vs. 27.4 months for the reverse sequence (p = 0.19).” The results are so markedly different for the ARM/BRCA1/BRCA2 group, it would be helpful to try to determine if this due to some imbalance in prognostic factors, ie prior taxane use in mCSPC, or volume of disease, etc.
  13. Lines 244-246: “Patients with somatic mutations of ATM/BRCA1/BRCA2 genes receive greater benefit from the standard mCRPC treatments than other patients, particularly ATM/BRCA1/BRCA2-mutated patients treated with taxanes”. I would soften this statement.  I would state explicitly "in our study these patients achieved longer PFS, etc".  I would also expect some statement on the limitations of the study, including retrospective nature, small sample size, heterogenous prior treatments, etc.

Author Response

Dear reviewer,

Thank you for your constructive remarks.

You find all point-by-point responses below.

Best regards,

Dr Zoé NEVIERE

  1. Line 46: “patients with DDR mutations are known to have a high response to PARP (Poly(ADP Ribose) Polymerase) inhibitors” - I would soften this statement. There is strong evidence that BRCA2 has high response rates to PARP inhibitors, but the other DDR genes appear to be far less responsive.

>> We have suppressed “ high” to soften this sentence.

  1. Minor comment: The use of the term new-generation hormone therapy (NGHT) is not consistent with other authors in the field, and these agents are no longer “new”.  The most frequently used terms are “androgen receptor pathway inhibitors (ARPIs)” and “novel hormonal therapies (NHT)”.  Perhaps consider changing to one of these terms.

>> We have changed NGHT to NHT in all the manuscript.

  1. Lines 50-52: “However, the debate remains open 50 since mutations were linked to better or worse survival outcomes after NGHT and/or taxane treatments”. I find this sentence structure awkward.  I may suggest changing to say that studies have shown conflicting results with taxanes and/or ARPIs.

>> We have changed the sentence “However, results of the different studies remains conflicting and linked to better or worse survival outcomes after NGHT and/or taxane treatments » to “However, results of the different studies remains conflicting about links between DDR mutations and survival outcomes after NGHT and/or taxane treatments”.

  1. Minor comment: Line 72 “WHO>2” Please define the “WHO” before using the acronym, and presumably this should read <2 (ie 0 or 1).

>> We have defined and WHO by World Health Organization (WHO) and changed > to < as evoked.

  1. Figure 1: Statement that 2 patients were excluded due to refusal.  Does this imply that the other patients provided consent?  If so I would include this in the methods, as this would strengthen the study.  I highlight this as it is not typical for retrospective studies to require patient consent, but is obviously ethically favourable if consent is obtained.

>> The alive patients received a non-opposition information note, it’s not a consent, in accordance with French law.

  1. Line 88: Please define what you mean by progression (ie biochemical and/or radiographic? and what criteria were used, ie PCWG3, RECIST, etc.).

>> We have added precision about progression: “biochemical as defined by French Association of Urology and/or radiologic progression according to PCWG3 and/or RECIST 1.1 criteria)”

  1. Line 122-123: “Thirty-three patients (39.8%) presented the DDR 122 mutation (DDR+)” Awkward sentence. Suggest changing to “Thirty-three patients (39.8%) presented the DDR 122 mutation (DDR+)”.

>> We have this sentence to “Thirty-three patients (39.8%) presented the somatic DDR mutation (they represent the DDR+ group).”

  1. Table 1: Please include units of measure in brackets beside each row title ex.  “Age (years)”.
  2. Table 1: Consider changing title “Duration of hormonosensitivity” to “time to castration resistance”.  Also, I assume this is measure in years?  See point above.

>> We have added a measure for age, duration of hormonsensititvity and time to castration resistance.

  1. Table 2: Suggest using accepted terminology of "pathogenic" and "likely pathogenic”

>> We have changed the terminology as suggested.

  1. Results section: For patients treated with taxanes, It is not clearly stated whether or not patients that received taxane chemotherapy in the mCSPC state are including in this analysis.  This is essential to know as one would expect lower response rates and shorter time to progression with retreatment.  It would be preferable to exclude these patients from the analysis (or do another analysis) to allow a more pure look at outcome from initial taxane use in this patient population.

>> Among patients who received taxanes in first line mCRPC, only one patient received taxanes before mCRPC, with a wash-out of 5 years. So we have conserved this patient.

  1. Lines 231-235: “In this chemotherapy and NGHT group, the 6 ATM/BRCA1/BRCA2 -231 mutated patients had a much longer PFS2 compared to patients with another or no mutation (median PFS2 of 35.7 months versus 7 months; p = 0.004). In ATM/BRCA1/BRCA2 mutated patients treated by taxane and then the NGHT sequence, PFS2 was particularly long (median PFS = 49.8 months) vs. 27.4 months for the reverse sequence (p = 0.19).” The results are so markedly different for the ARM/BRCA1/BRCA2 group, it would be helpful to try to determine if this due to some imbalance in prognostic factors, ie prior taxane use in mCSPC, or volume of disease, etc.

>>We have compared the 6 ATM/BRCA1/BRCA2-mutated patients to the 32 others patients, and there is no statistical difference about age, Gleason, initial PSA, metastasis at diagnosis and other characteristic reported in the Table 1.

  1. Lines 244-246: “Patients with somatic mutations of ATM/BRCA1/BRCA2 genes receive greater benefit from the standard mCRPC treatments than other patients, particularly ATM/BRCA1/BRCA2-mutated patients treated with taxanes”. I would soften this statement.  I would state explicitly "in our study these patients achieved longer PFS, etc".  I would also expect some statement on the limitations of the study, including retrospective nature, small sample size, heterogenous prior treatments, etc.

>> We have changed the sentence as suggested to “In our study, patients with somatic mutations of ATM/BRCA1/BRCA2 genes achieved longer PFS with standard mCRPC treatments than other patients. They seems to receive greater benefit from taxanes

and added a sentence to focus on our limits at the end of the discussion.”.We have added a sentence at the beginning of the discussion :” Because this study presents several limitations such as monocentric and retrospective characteristics, small size sample and heterogenous population (prior treatment, metasis at diagnosis…), new prospective studies with more homogenous patients would be needed to confirm these results.

Round 2

Reviewer 1 Report

The authors replied sufficiently to my comments

Reviewer 2 Report

My comments have been addressed to my satisfaction.